Article 

# C(*alkyl*)−C(*vinyl*) bond cleavage enabled by Retro-Pallada-Diels-Alder reaction

Qingyang Zhao [1,2,3], Le Yu[1,3], Yao-Du Zhang[1,3], Yong-Qiang Guo[1], Ming Chen[1], Zhi-Hui Ren[1] & Zheng-Hui Guan [1] ✉

Activation and cleavage of carbon−carbon (C−C) bonds is a fundamental transformation in organic chemistry while inert C−C bonds cleavage remains a long-standing challenge. Retro-Diels-Alder (retro-DA) reaction is a well-known and important tool for C−C bonds cleavage but less been explored in methodology by contrast to other strategies. Herein, we report a selective C(*alkyl*)−C(*vinyl*) bond cleavage strategy realized through the transient directing group mediated retro-Diels-Alder reaction of a six-membered palladacycle, which is obtained from an in situ generated hydrazone and palladium hydride species. This unprecedented strategy exhibits good tolerances and thus offers new opportunities for late-stage modifications of complex molecules. DFT calculations revealed that an intriguing retro-Pd(IV)-Diels-Alder process is possibly involved in the catalytic cycle, thus bridging both Retro-Diels-Alder reaction and C−C bond cleavage. We anticipate that this strategy should prove instrumental for potential applications to achieve the modification of functional organic skeletons in synthetic chemistry and other fields involving in molecular editing.

Carbon-carbon (C−C) bonds cleavage is a fundamental and valuable transformation in organic and material sciences for rapid construction and modification of functional organic skeletons[1–3]. However, compared with the significant progress in C−C bonds formations, extensive efforts in this field have always been hampered by the inherent stability of C−C linkages. Up to date, several metal-mediated strategies, such as oxidative addition[4–10], β-carbon elimination[11–14], retro-allylation[15] and decarboxylation[16], have rendered the cleavage of strained and polar C−C bonds feasible. Nevertheless, novel strategies for selective cleavage of inert (unstrained and nonpolar) C-C bonds are still in great demand in both academia and industry[17–22].

Retro-Diels-Alder (retro-DA) reaction is a well-known and important tool for C−C bonds cleavage[23–25], which usually breaks two δ bonds and forms three π bonds simultaneously. Up to now, retro-DA reaction has been used as the key reaction in the synthesis of compounds difficult to obtain by other methods, including polymers, macromolecules, complex natural products and bioactive molecules[26,27].

Notwithstanding, the capacity of retro-DA reaction has not yet been as fully reached as that of Diels-Alder reaction. The pioneering research of the metalla-Diels-Alder reaction was reported by Wulff in 1989 with a chromadiene as the 4π partner, whereas a terminal carbon in the diene was replaced by chromium (Fig. 1a)[28,29]. Afterwards, Trost[30] and Reißig[31] reported the elegant pallada-DA and rhoda-DA reactions. According to the principle of microscopic reversibility, retro-metalla-DA reaction should be applicable as a novel strategy to cleave C−C bonds in theory;[32] the formal similarity between the six-membered metallacycles and the six-membered cyclic transition state in retro-DA reactions enhances the feasibility of this method. Furthermore, the universality of cyclometallation makes this strategy a potential complementary and alternative solution to C−C bonds cleavage. However, to the best of our knowledge, there is no report of C−C bond cleavage based on retro-metalla-DA reactions. The main challenge lies in the transformation between carbon−metal (C−M) bonds and C−C bonds: (1) formation of six-membered metallacycle is thought to be thermodynamically less

[1]Key Laboratory of Synthetic and Nature Functional Molecule of the Ministry of Education, Department of Chemistry & Materials Science, Northwest University, Xi'an, P.R. China. [2]School of Pharmaceutical Sciences (Shenzhen), Shenzhen Campus of Sun Yat-sen University, Shenzhen, P.R. China. [3]These authors contributed equally: Qingyang Zhao, Le Yu, Yao-Du Zhang. ✉e-mail: guanzhh@nwu.edu.cn

a)

b)

**Fig. 1 | C-C bond cleavage involving retro-DA reaction. a** Retro-Metalla-Diels-Alder (DA) reaction for C−C bonds cleavage. **b** Retro-Pallada-DA reaction for C(*alkyl*)-C(*vinyl*) bonds cleavage.

## Table 1 | Optimization of Retro-Pallada-DA reaction[a]

| Entry | R | [Pd] | Additive | Solvent | Yield (%) | | |
|---|---|---|---|---|---|---|---|
| | | | | | 2a | 3a | 4a |
| 1 | -Ts | Pd(PPh₃)₄ | HBr | Tol | 3 | 5 | 6 |
| 2 | -Ac | Pd(PPh₃)₄ | HBr | Tol | 16 | 19 | 20 |
| 3 | -ᵗBu | Pd(PPh₃)₄ | HBr | Tol | 15 | 11 | 9 |
| 4 | -Bz | Pd(PPh₃)₄ | HBr | Tol | 11 | 15 | 16 |
| 5 | -Ac | PdBr₂ | — | Tol/ⁱPrOH | 23 | 12 | 8 |
| 6 | -Ac | PdBr₂ | PPh₃ | Tol/ⁱPrOH | 20 | 0 | 4 |
| 7 | -Ac | PdBr₂ | XantPhos | Tol/ⁱPrOH | 18 | 5 | 3 |
| 8 | -Ac | PdBr₂ | CO | Tol/ⁱPrOH | 35 | 9 | 2 |
| 9 | -Ac | PdBr₂ | CO | DCE/ⁱPrOH | 61 | 0 | 0 |
| 10[b] | -Ac | PdBr₂ | CO | DCE/ⁱPrOH | <5 | 0 | 0 |
| 11 | -Ac | PdBr₂ | CO/O₂ | DCE/ⁱPrOH | 83 | 0 | 0 |
| 12[c] | -Ac | PdBr₂ | CO/O₂ | DCE/ⁱPrOH | 86(82) | 0 | 0 |
| 13 | -Ac | — | CO/O₂ | DCE/ⁱPrOH | 0 | 0 | 0 |

[a] Optimization studies; reactions were performed with 0.2 mmol substrates (hydrazone or ketone) in 2.0 mL of solvent for 24 h; GC-FID yield was given. [b] The reaction was operated in a glovebox in the absence of oxygen. [c] 1-Phenylbut−3-en-1-one **1a** and acetohydrazide were used as the substrates; isolated yield in parenthesis. Ts p-toluenesulfonyl, Ac acetyl, Bz benzoyl.

favored than its five-membered counterparts[33]; (2) several side reactions on the six-membered metallacycles, such as reductive elimination[34] and β-hydrogen elimination[35], are more facile than the retro-metalla-DA reaction; (3) the harsh conditions that are generally required to drive the retro-DA reactions might lead to the side

reactions more prominently[25]. To this end, it is attractive but challenging to address the gap in retro-metalla-DA reactions.

Owing to the increasingly extensive and insightful investigations into organopalladium over the last few decades[36,37], we envision that a significant breakthrough might be made in the field of C−C bonds

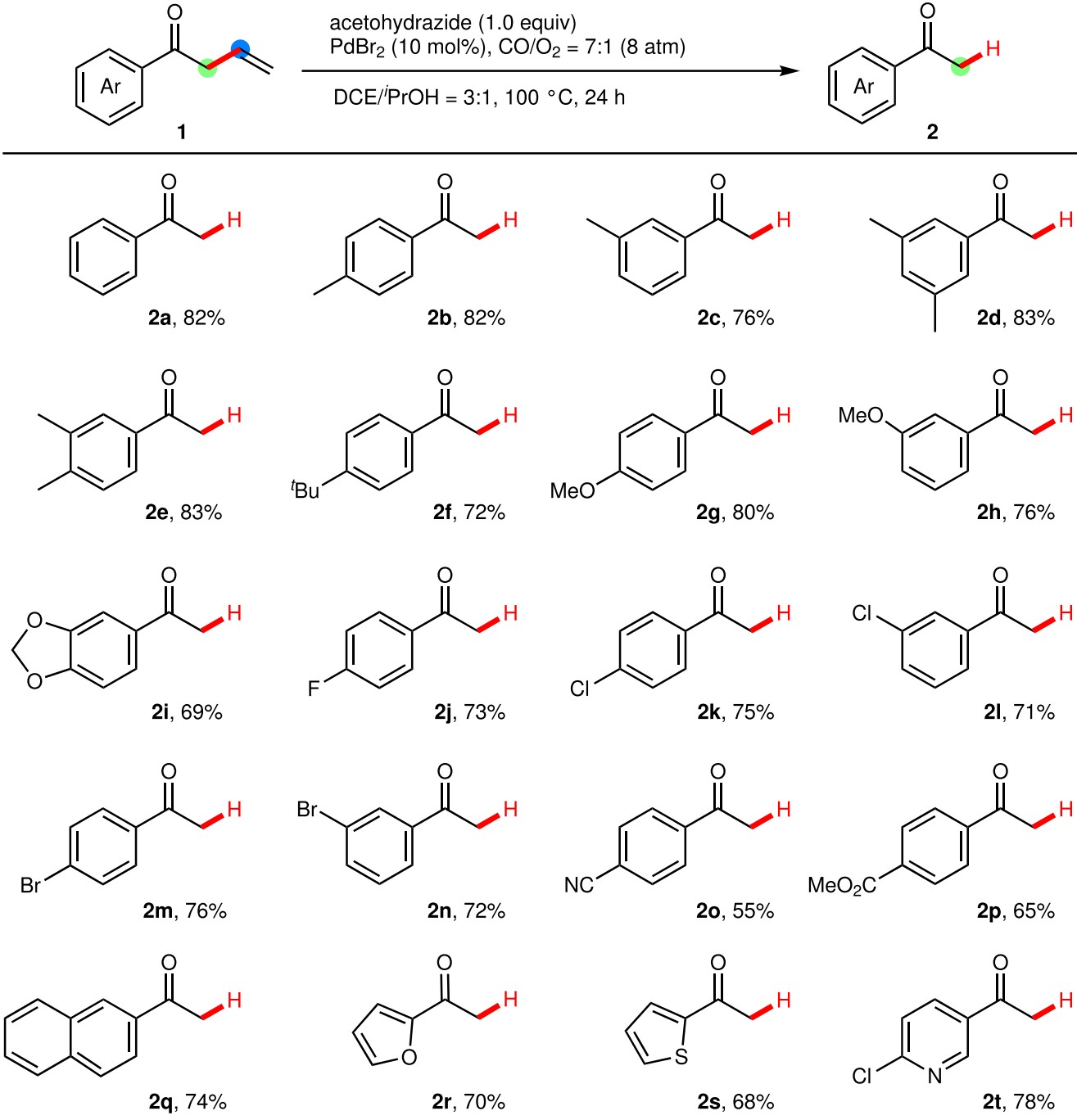

**Fig. 2 | Scope of substituents on (het)aryl groups (R). a** Conditions: Ketone 1 (0.2 mmol), acetohydrazide (0.2 mmol), PdBr$_2$ (10 mol%), CO/O$_2$ = 7:1 (8 atm), DCE/$^i$PrOH = 3:1 (2.0 mL), 100 °C for 24 h. Isolated yield.

cleavage through the rational design of palladacycles[38]. As shown in Fig. 1b, a six-membered hydrazone-based palladacycle was designed as an ideal substrate for the following reasons: (1) hydrazine-based keti-mine has been used for generating palladacycles[39]; (2) azoalkene has been used as a diene in DA reactions[40]; (3) it is convenient to directly observe the cleavage selectivity of different types of C−C bonds. According to this hypothesis, the cleavage would selectively occur at C(*alkyl*)-C(*vinyl*) bond via a retro-pallada-DA reaction, even in the presence of more active bonds such as polar C(*imino*)-C(*alkyl*) and various C−H bonds.

Here, we show a selective C(*alkyl*)−C(*vinyl*) bond cleavage strat-egy through the retro-metalla-Diels-Alder reaction of a six-membered

hydrazone-based palladacycle, in situ generated from ketone, hydra-zide and palladium catalyst.

## Results and discussion

Initially, hydropalladation of α-allyl ketimine (**1 A**) was utilized to generate the hypothetic six-membered palladacycle and then perform the transformation (Table 1). Different hydrazones were screened in the presence of palladium catalysts and acid additives. After many attempts, the desired C−C bond cleavage was indeed observed in the presence of Pd(PPh$_3$)$_4$ and HBr in toluene (Table 1, entries 1–4). It was found that the *N*-Ac and *N*-$^t$Bu hydrazones showed better reactivity than *N*-Ts and *N*-Bz hydrazones, giving acetophenone **2a** directly with

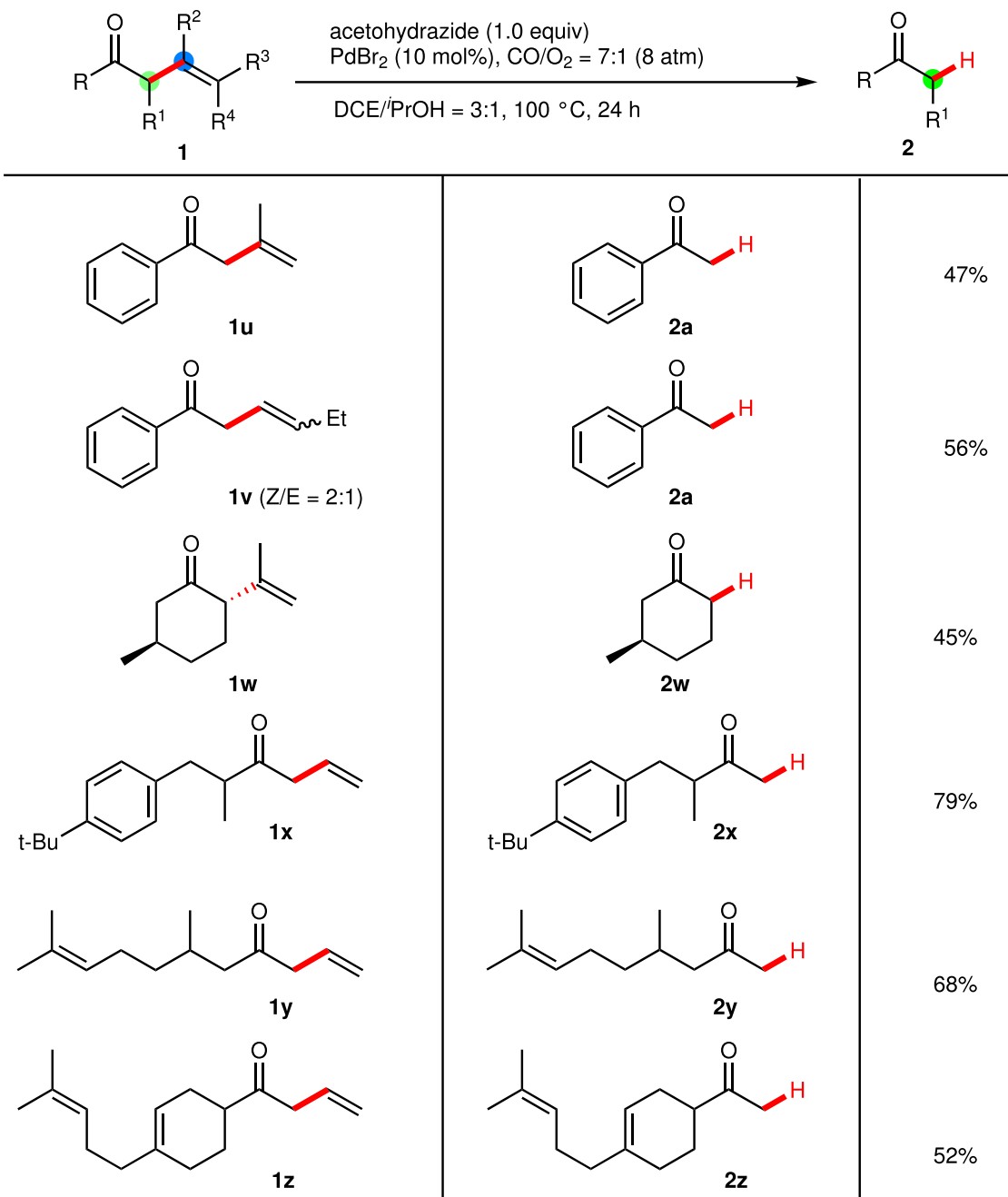

**Fig. 3 | Scope of substituents on allyl groups (R¹-R⁴) and alkyl/alkenyl groups (R). a** Conditions: Ketone 1 (0.2 mmol), acetohydrazide (0.2 mmol), PdBr$_2$ (10 mol%), CO/O$_2$ = 7:1 (8 atm), DCE/$^i$PrOH = 3:1 (2.0 mL), 100 °C for 24 h. Isolated yield.

15 and 16% yields, although predictable byproducts **3a** and **4a** were also detected. These preliminary results verified that our strategy was feasible in certain cases.

Therefore, *N*-Ac hydrazone was chosen as the standard substrate for improving the reaction efficiency. Further studies suggested that the palladium hydride in situ generated from PdBr$_2$/2-propanol displayed higher efficiency (Table 1, entry 5–9). Probably because of the promoting effects of CO in the formation of palladium hydride catalyst[41–43], an important improvement was observed in the presence of CO, affording acetophenone **2a** with up to a 61% yield in DCE (Table 1, entry 9). Furthermore, we found that the traces of oxygen played an important role in the reaction (Table 1, entry 10 *vs.* 11). Through an extensive screen of the conditions, the yield of **2a** was greatly improved from 61 to 83% in the presence of 1 atm O$_2$ (Table 1, entry 11). Considering that ketone is more common than hydrazone in

organic materials, as well as the broad utility of hydrazine as a traceless directing group in the transformation of ketones, we tried the one-pot procedure by using 1-phenylbut-3-en-1-one **1a** and acetohydrazide as the substrates. To our delight, an 82% isolated yield of acetophenone **2a** was obtained in this case (Table 1, entry 12). For comparison, the reaction did not occur in the absence of palladium catalyst (Table 1, entry 13).

Under optimized reaction conditions, the one-pot procedure was carried out for the structurally diverse screening of substrates. As shown in Fig. 2, this catalytic system is of high functional group tolerance. Aryl allyl ketones possessing *para-* or *meta-* electron-donating groups on the aryl ring, such as alkyl, methoxy and [1,3]dioxole, underwent the C–C bond cleavage reaction easily to afford the corresponding aryl methyl ketones (**2b-2i**) with 69–83% yields. Aryl allyl ketones with halide substituents (F, Cl and Br), which were potentially

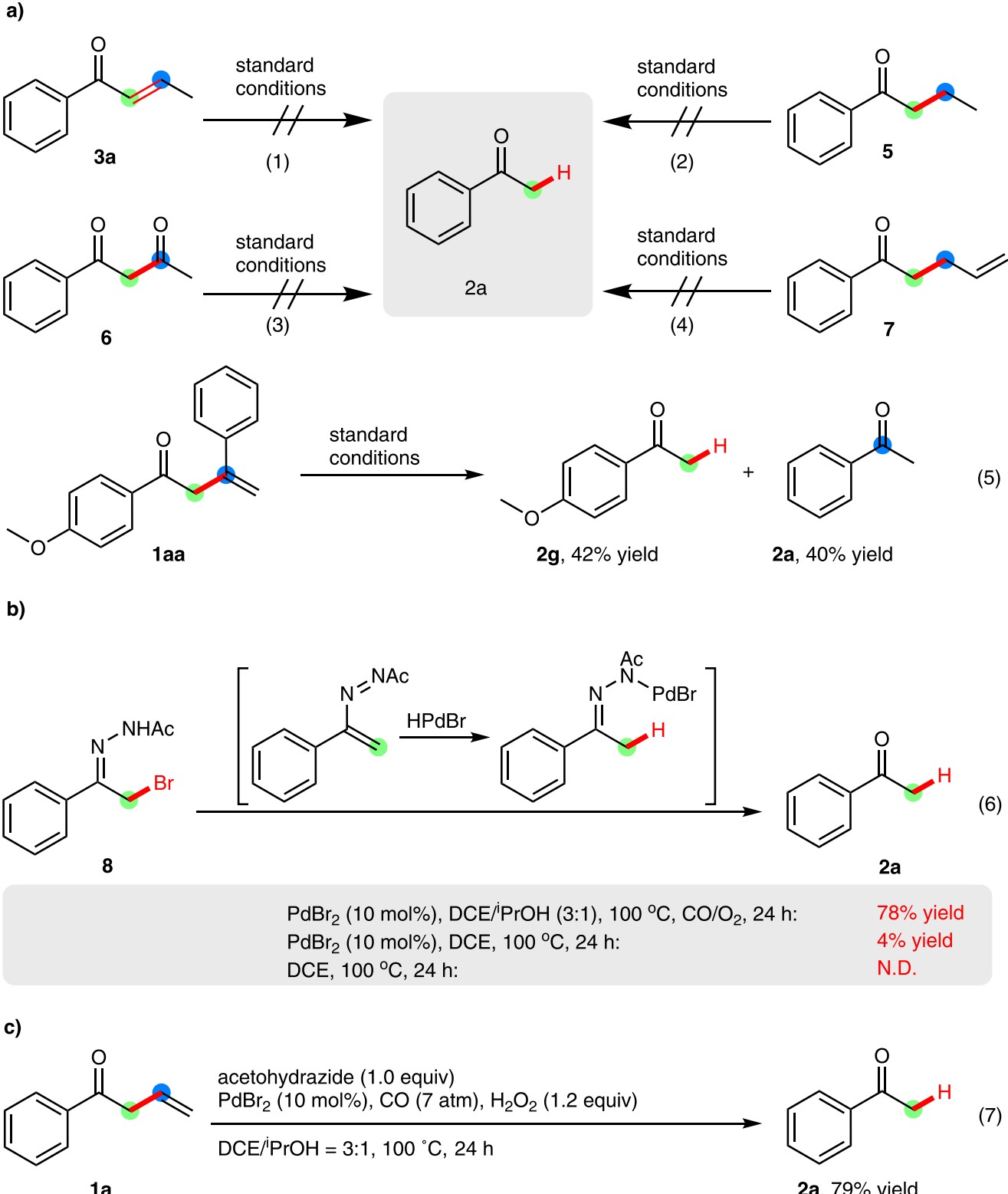

**Fig. 4 | Control Experiments. a** Investigation of potential pathway and key intermediates by using substituted substrates. **b** Investigation of potential intermediates by using azoalkene precursor. **c** Investigation of potential role of $O_2$ by using $H_2O_2$.

useful in cross-coupling reactions, were compatible with the optimized conditions to produce the desired products in good to high yields (**2j-2n**). Notably, strong electron-withdrawing functional groups, such as cyano and ester, afforded the desired acetophenones with satisfiing yields (**2o** and **2p**). Besides, this novel catalytic system was applicable for naphthyl and heteroaryl allylic ketones. Representative

2-naphthyl methyl ketone **2q** and heteroaryl methyl ketones **2r-2t** were obtained in 68–78% yields.

For investigating the universality and regioselectivity of this novel strategy, substrates with substituted vinyl groups were investigated (Fig. 3). Substrate **1u** gave **2a** in moderate yield, probably because of the steric effect of the methyl group on the C(β). A similar

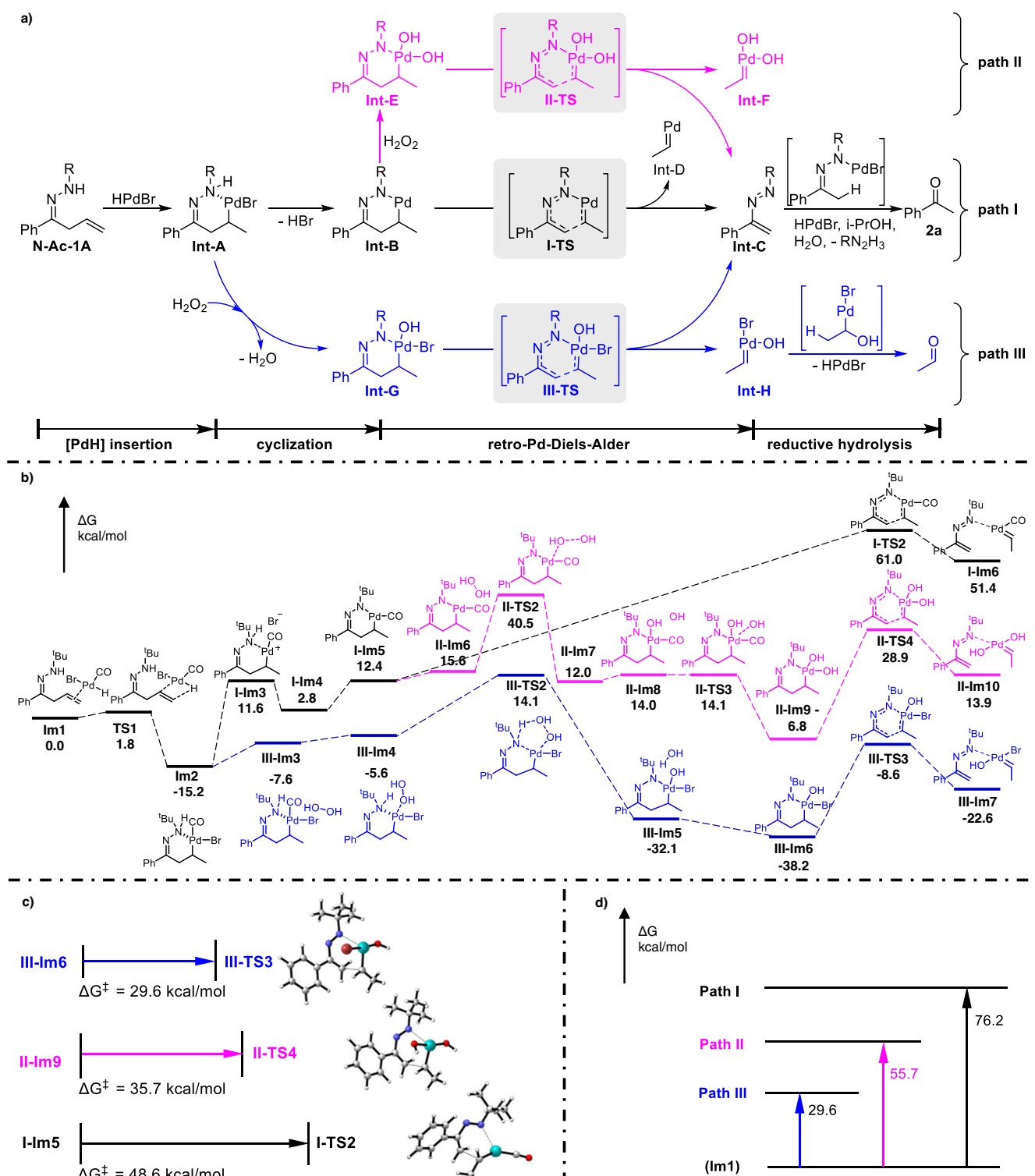

**Fig. 5 | Mechanism investigation. a** Proposed Mechanism (ligands are omitted for clarity). **b** DFT calculated relative Gibbs free energy profiles for the **paths I, II** and **III** by B3LYP-D3(dichloroethane)/{6-311 + +G(d,p), SDD(Pd)} with respective to **Im1**. **c** Activation barrier of retro-Pd-DA step. **d** Activation barrier of the overall path.

phenomenon was observed for C(γ)-ethyl substituted substrate **1 v**, whereas the coordination of the vinyl groups to the metal center was probably suppressed due to the steric effects. Furthermore, alkyl allylic ketones were screened under standard conditions. It is noted that **2w** could be obtained from nature product, C(α)- and C(β)-di-substituted (+)-isopulegone **1w**, in a 45% yield. The **2x** was also obtained in a 79% yield when alkyl allylic ketone **1x** was used as the

substrate. Furtherly, multiple vinyl substituted substrates with the vinyl group at different positions were tested in the reaction (**1 y** and **1z**). Remarkably, both cyclic C(γ)-vinyl and remote vinyl groups were well retained to afford the desired products **2y-2z** in moderate to good yields, demonstrating the chemo- and regioselectivity of the cleavage strategy and potential further application for late-stage synthetic modifications of complex molecules.

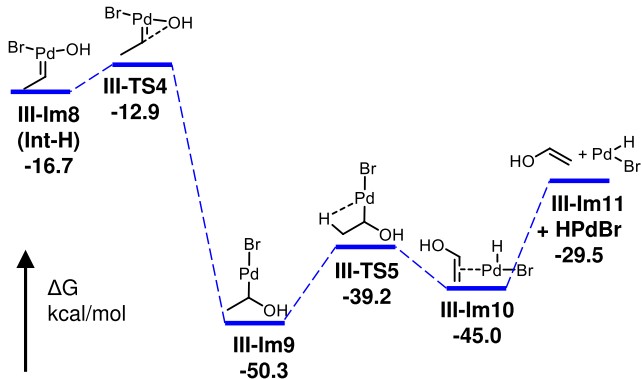

**Fig. 6 | DFT calculated Gibbs free energy profiles for regeneration HPdBr catalyst from Int-H.** Origin of the free energies is with respect to **Im1** in Fig. 5b.

To gain insights into this novel C–C bond cleavage process, a series of detailed control experimental studies were performed under the standard conditions (Fig. 4). The migration (**3a** and **7**), hydrogenation (**5**) or oxygenation (**6**) of the vinyl group resulted in the failure of obtaining desired product **2a**, implying a remarkable dependency of β,γ-vinyl ketone conformation and highly specific selectivity of the reaction site. These results also excluded the potential participation of **3a**, **5** or **6** as the key intermediates. Furthermore, a piece of important experimental evidence was observed when 1-(4-methoxyphenyl)-3-phenylbut-3-en-1-one **1aa** was used as the substrate under the standard conditions, wherein the desired ketone **2 g** was obtained in 42% yield accompanied with ketone **2a** in 40% yield (Fig. 4, equation 5). Afterwards, *N*'-(2-bromo-1-phenylethylidene) acetohydrazone **8**, a precursor of 'diene' azoalkene[44–48], gave **2a** in a 78% yield under standard conditions (Fig. 4, equation 6), demonstrating the possibility that azoalkene was involved as an intermediate in the reaction. In addition, considering the critical role of PdBr₂ and 2-propanol in this transformation, we assumed that the final ketone **2a** might be produced through a process of 1,4-addition of palladium-hydride into azoalkene species followed by hydrolysis (see Supplementary Fig. 3).

Based on the above results, we proposed that this C(*alkyl*)–C(*vinyl*) bond cleavage proceeded through the following key steps involving a six-membered palladacycle (Fig. 5a, **path I**): (1) insertion of palladium-hydride species (HPdBr) to **1 A** generate an alkyl-palladium intermediate **Int-A**;[49–51] (2) cyclization of the **Int-A** forms a six-membered palladacycle **Int-B**; (3) retro-Pd(II)-DA reaction of the palladacycle affords an azo-'diene' **Int-C** and a Pd-carbene; and (4) reductive hydrolysis of **Int-C** gives the final product **2a**. Detailed processes were explored by carrying on DFT calculations (Fig. 5b). The *N*-ᵗBu-**1A**, which has similar reactivity as *N*-Ac-**A** under the standard conditions, was used as the model substrate to restrain the intramolecular coordination (see Supplementary Information).

For the **path I** (Fig. 5b, black line), the steps (1) and (2) seem to be reasonable with an overall activation barrier of 27.6 kcal/mol, while the individual retro-Pd(II)-DA step requires a relatively high barrier of 48.6 kcal/mol with respect to **I-Im5** (Fig. 5c). However, the transformation from **Im1** to **I-Im6** requires an overall activation barrier of 76.2 kcal/mol (Fig. 5d, **path I**), which is highly prohibited in theory. Considering that oxygen played an important role in the reaction (Table 1, entry 10), as well as that the reaction proceeded smoothly to give the desired product **2a** in 79% yield when 1.2 equiv. of hydrogen peroxide was used instead of oxygen (Fig. 4, equation 7), we hypothesized that the hydrogen peroxide would be easily formed from O₂/DCE/ᵢPrOH in the presence of palladium catalyst and improve the transformation[52,53]. Therefore, two competitive mechanistic pathways are proposed as an alternative, wherein the Pd(IV) intermediates might

be involved in the reaction[54]. In this context, oxidation of Pd(II) to Pd(IV) may take place at **Int-A** or **Int-B**, namely as Pd(II) cyclization-retro-Pd(IV)-DA (Fig. 5a, **path II**) and Pd(IV) cyclization-retro-Pd(IV)-DA (Fig. 5a, **path III**).

For the **path II** (Fig. 5b, purple line), six-membered Pd(IV)-cycle **II-Im9** is formed from **Im1** with an overall barrier of 55.7 kcal/mol through the sequential insertion, cyclization and oxidation. Afterwards, the retro-Pd(IV)-DA reaction of **II-Im9** proceeds with an activation barrier of 35.7 kcal/mol to give **II-Im10** via transition state **II-TS4** (Fig. 5c). As a result, the transformation from **Im1** to **II-Im10** requires a relatively high overall activation energy of 55.7 kcal/mol (Fig. 5d, **path II**).

For the **path III** (Fig. 5b, blue line), six-membered Pd(IV)-cycle **III-Im6** is formed from **Im1** with an overall activation barrier of 29.3 kcal/mol through sequential insertion of palladium-hydride (**I-Im2**), binding with H₂O₂ (**III-Im3**), releasing of CO (**III-Im4**) and H₂O (**III-Im5**). Then, the retro-Pd(IV)-DA reaction of **III-Im6** occurs via transition state **III-TS3** with an activation barrier of 29.6 kcal/mol, which is more energetically favorable than **path I** and **path II** (Fig. 5c). As a result, the transformation from **Im1** to **III-Im7** requires the lowest overall activation barries of three pathways (29.6 *vs* 55.7 and 76.2 kcal/mol, respectively) (Fig. 5d, **path III**). Subsequently, the exothermic OH migration of **III-Im8** requires lower barriers and provides driving forces to the retro-Pd(IV)-DA step (Fig. 6)[55]. Notably, the DFT calculations indicated that CO played an important role in the transformation of the active palladium hydride species[41–43], which is consistent with the experimental results in this work.

In summary, we have reported an unconventional retro-pallada-Diels-Alder reaction for the first time and realized the palladium catalyzed inert C(*alkyl*)–C(*vinyl*) bonds cleavage via this strategy. The key six-membered Pd(IV)-cycle is probably generated from the sequential hydrazide-assisted [PdH] insertion into the C=C double bond and oxidation of the Pd(II) intermediate, which is supported by our experimental and computational investigations. This strategy has achieved highly selective transformation of a series of substituted substrates. Therefore, we anticipate that this unprecedented tactic would offer new opportunities for inert C-C bond cleavage, and therefore potentially applicable to achieve the modification of functional organic skeletons.

## Methods

### Typical procedure for palladium-catalyzed C(*alkyl*)-C(*vinyl*) cleavage

Ketone **1a** (0.2 mmol), PdBr₂ (10 mol%, 5.3 mg), acetohydrazide **2a** (0.2 mmol, 14.8 mg) and 2 mL mixed solvents (DCE/ᵢPrOH = 3/1) was added to a 10 mL glass vial capped with perforated aluminum foil. The vial was transferred into an autoclave, which was carefully evacuated and backfilled with O₂ (1 atm) and then filled by CO (7 atm). After being stirred at 100 °C oil bath for 24 h, the autoclave was cooled down to room temperature and then the gas was released slowly. The reaction mixture was quenched with H₂O (10 mL) and extracted with EtOAc (3 × 10 mL). The combined organic layers were dried over anhydrous Na₂SO₄ and then evaporated in vacuo. The residue was purified by column chromatography on silica gel to afford the corresponding acetophenone **2a** with hexanes/EtOAc (20/1) as the eluent.

## Data availability

The authors declare that the data relating to the characterization of products, experimental protocols and the computational studies are available within the article and its Supplementary Information.

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

## Acknowledgements

We thank the financial support provided by the National Natural Science Foundation of China (21971204, 22171224), the Innovation Capability Support Program of Shaanxi Province (No. 2020TD-022) and Fund of Education Department of Shaanxi Provincial Government (22JP082). Q.Z. thanks the Science, Technology and Innovation Commission of Shenzhen (JCYJ20190807155201669).

## Author contributions

Z.G. conceived the original idea and directed the research. L.Y. performed the DFT calculations. Z.G. L.Y. and Q.Z. designed experiments and analyzed the data. Y.Z., Y.G., M.C., Z.R. performed the synthetic experiments. Q.Z., L.Y., and Z.G. wrote the manuscript with feedback from all the authors.

## Competing interests

The authors declare no competing interests.
