## [Peer Review File · Nature Communications]

REVIEWER COMMENTS

Reviewer #1 (Remarks to the Author):

The manuscript of "C(alkyl)-C(vinyl) Bond Cleavage Enabled by Retro-Pallada-Diels-Alder Reaction" by Guan and co-workers is conceptually interesting. It is well known that selective cleavage of C-C bonds is a fundamental transformation in chemical synthesis. In the past several years, excellent works on this topic have been developed. However, the vast majority of the reported works are based on just several fundamental strategies, such as oxidative addition, β -carbon elimination, retro-allylation and decarboxylation, new strategy for cleavage of inert C-C bonds is still very rare and remains highly desirable.

In this manuscript, the authors report that α -allyl ketimines undergo a C-C bond cleavage reaction by palladium catalysis through an unprecedented retro-Pd-DA mechanism. Under the standard conditions, a series of substrates in Table 2 and Table 3 were tolerated to afford the corresponding C(alkyl)-C(vinyl) bonds cleavage products in good to high yields. The control experiments in Scheme 2 have confirmed the proposed retro-Pd-DA mechanism. They have also done a detailed DFT calculations to gain insight into the exact mechanism of the reaction. The DFT calculations was a high quality study and proved that the Pd(IV) cyclization-retro-Pd(IV)-DA (Figure 1A, path III) was the much reasonable process. Notably, when oxygen was instead by hydrogen peroxide, the desired product 2a was obtained in 79% yield in Scheme 2 Eq. (7). The results of computational chemistry are fully consistent with the experimental results. Thus, the Pd(IV) cyclization-retro-Pd(IV)-DA pathway is fully convinced. Another advance of the paper is of that the role of CO in the reaction (facilitate formation of active palladium hydride species) has been well-reasoned by computational chemistry.

Overall, the results in this manuscript are highly important, the work has been carried out with high technical quality and the paper has significant depth and novelty. I think this is a nice paper that deserves publication in Nature Communications.

The following small problems need attention:

1. In order to facilitate readers to repeat DFT calculation, please show the imaginary frequencies of all transition states in the supporting information.
2. On the page 2, introduction, the second paragraph. The retro D-A reaction has long been a difficult problem and has not been solved. This is because it is inverse thermodynamics. In order to solve this problem, the current method is to make the retro-D-A product form aromaticity, thus decreases the Gibbs free energy of the product (J. Am. Chem. Soc. 2019, 141, 9731-9738, Angew. Chem. Int. Ed. 2017, 56, 7166). There is no direct solution to this problem. Therefore, any form of retro D-A reaction is important and challenging. Please amend the description of this point in this paragraph.
3. Some recent papers should be cited:

(1) β -Carbon Elimination: Nature Chemistry 2022, 14, 398-406;

(2) Oxidative Cleavage: Nature 2022, 610, 81-86.

(3) Dyotropic Rearrangement: J. Am. Chem. Soc. 2022, 144, 14047-14052.

4. In abstract, this novel strategy is pioneering research of unprecedented retro-metalla-DA reaction. As a new type of retro-DA reaction, "retro-pallada-Diels-Alder" seems to be better than "retro-pallada-(hetero)DA reaction".

Reviewer #3 (Remarks to the Author):

This paper described an interesting C–C bond cleavage protocol through retro-pallada-Diels-Alder pathway. The reaction shows promising application prospect in chemical modification of β,γ -vinyl ketones due to its high site-selectivity and good functional group tolerances. The authors have proposed six-membered palladacycle mechanisms, and have investigated the detailed mechanism by control experiments and DFT calculations.

After carefully checking the DFT calculations in the paper, this reviewer finds that the selected computation methods are appropriate and provides reliable theoretical insights for this complex chemical synthetic experiment. Step-by-step mechanism presented in this paper suggests that the energetically favored path-III, that well interprets the Pd-catalyzed cyclization and ring-opening processes. The promotion effects of CO and O₂ in their experiments have been reasonably explained through these computational results.

In my opinion, the elaborate reaction path network proposed in this paper contributes to get deeper understanding of the retro-pallada-DA reactions and offers valuable reference for future experimental explorations. I recommend acceptance after addressing the following minor comments.

1) The zero energy should be addressed in SI. The relative energies and free energies (at 298.15K) with respect to Im1 are in kcal/mol.

2) ΔG_{TS-a} in Figure S1 was relative difference between ΔG of Im-a and TS-a, which is better expressed as $\Delta\Delta G$.

3) Fonts in Table S1 is not consistent.

Reviewer #2 (Remarks to the Author):

These authors reported a palladium-catalyzed reductive cleavage of a C(alkenyl)-C(alkyl) bond. The authors proposed that the beta-carbon elimination reaction takes place via retro-Metala-DA reaction based on the experimental results and theoretical calculations. However, this reviewer is skeptical of their proposed mechanism or working hypothesis of this reaction. Their results shown in equations 5 and 7 can be explained by the following conventional reaction pathway (see the scheme below). Electrophilic palladium(II) species can bind to the alkene moiety of the ketone and then the nucleophilic attack of water generated by a condensation reaction of the ketone with acetohydrazine to the coordinated alkene to give intermediate A in the following scheme. Subsequently, β -carbon elimination proceeds to give the corresponding alkyl palladium, oxa- π -allyl palladium, or palladium enolate complex B. Protonolysis of the palladium with generated HX to give the corresponding ketones and regenerate palladium(II) species.

The reaction mechanism claimed by the authors is speculative, and their results are insufficient to explain their proposed mechanism. Based on the experimental results, they should exclude other possible conventional mechanisms such as those mentioned above.

They described their working hypothesis using metalacyclohexene in Scheme 1A, and their proposed intermediate is a metalateterahydropyridazine derivative, not a metalacyclohexene intermediate. This description needs to be more accurate to understand the novelty of this type of metala-electrocyclic reaction, and they should rationally describe the working hypothesis using a suitable intermediate.

As described above their result described in this manuscript does not satisfy the requirement of Nature Communications.

[Additional comments]

On page 3, line 52 and 53: they should cite references for supporting the contents (2) and (3).

Wrong nomenclature: isopropanol \rightarrow isopropyl alcohol or 2-propanol

Point-by-point response to the reviewers' comments

To Reviewer 1:

The manuscript of “C(alkyl)–C(vinyl) Bond Cleavage Enabled by Retro-Pallada-Diels-Alder Reaction” by Guan and co-workers is conceptually interesting. It is well known that selective cleavage of C-C bonds is a fundamental transformation in chemical synthesis. In the past several years, excellent works on this topic have been developed. However, the vast majority of the reported works are based on just several fundamental strategies, such as oxidative addition, β -carbon elimination, retro-allylation and decarboxylation, new strategy for cleavage of inert C-C bonds is still very rare and remains highly desirable.

In this manuscript, the authors report that α -allyl ketimines undergo a C-C bond cleavage reaction by palladium catalysis through an unprecedented retro-Pd-DA mechanism. Under the standard conditions, a series of substrates in Table 2 and Table 3 were tolerated to afford the corresponding C(alkyl)–C(vinyl) bonds cleavage products in good to high yields. The control experiments in Scheme 2 have confirmed the proposed retro-Pd-DA mechanism. They have also done a detailed DFT calculations to gain insight into the exact mechanism of the reaction. The DFT calculations was a high quality study and proved that the Pd(IV) cyclization-retro-Pd(IV)-DA (Figure 1A, path III) was the much reasonable process. Notably, when oxygen was instead by hydrogen peroxide, the desired product 2a was obtained in 79% yield in Scheme 2 Eq. (7). The results of computational chemistry are fully consistent with the experimental results. Thus, the Pd(IV) cyclization-retro-Pd(IV)-DA pathway is fully convinced. Another advance of the paper is of that the role of CO in the reaction (facilitate formation of active palladium hydride species) has been well-reasoned by computational chemistry.

Overall, the results in this manuscript are highly important, the work has been carried out with high technical quality and the paper has significant depth and novelty. I think this is a nice paper that deserves publication in Nature Communications. The following small problems need attention:

1. In order to facilitate readers to repeat DFT calculation, please show the imaginary frequencies of all transition states in the supporting information.

Response: Thanks. The imaginary frequencies have been listed in the revised SI.

2. On the page 2, introduction, the second paragraph. The retro D-A reaction has long been a difficult problem and has not been solved. This is because it is inverse thermodynamics. In order to solve this problem, the current method is to make the retro-D-A product form aromaticity, thus decreases the Gibbs free energy of the product (J. Am. Chem. Soc. 2019, 141, 9731-9738, Angew. Chem. Int. Ed. 2017, 56, 7166). There is no direct solution to this problem. Therefore, any form of retro D-A reaction is important and challenging. Please amend the description of this point in this paragraph.

Response: Thanks. Yes, *any form of retro D-A reaction is important and challenging.* Aromaticity process is indeed a promising strategy to enable retro D-A reaction. We have revised our introduction and cited the papers in ref 26 and 27.

3. Some recent papers should be cited:

- (1) β -Carbon Elimination: Nature Chemistry 2022, 14, 398-406;
- (2) Oxidative Cleavage: Nature 2022, 610, 81-86.
- (3) Dyotropic Rearrangement: J. Am. Chem. Soc. 2022, 144, 14047-14052.

Response: Thanks. We cited these recent papers as reference 10, 14, and 17.

4. In abstract, this novel strategy is pioneering research of unprecedented retro-metalla-DA reaction. As a new type of retro-DA reaction, “retro-pallada-Diels-Alder” seems to be better than “retro-pallada-(hetero)DA reaction”.

Response: Thanks. According to reviewer’s kind comment, we revised our manuscript.

To the Reviewer 2:

1. These authors reported a palladium-catalyzed reductive cleavage of a C(alkenyl)-C(alkyl) bond. The authors proposed that the beta-carbon elimination reaction takes place via retro-Metala-DA reaction based on the experimental results and theoretical calculations. However, this reviewer is skeptical of their proposed mechanism or working hypothesis of this reaction. Their results shown in equations 5 and 7 can be explained by the following conventional reaction pathway (see the scheme below). Electrophilic palladium(II) species can bind to the alkene moiety of the ketone and then the nucleophilic attack of water generated by a condensation reaction of the ketone with acetohydrazine to the coordinated alkene to give intermediate A in the following scheme. Subsequently, beta-carbon elimination proceeds to give the corresponding alkyl palladium, oxa-pi-allyl palladium, or palladium enolate complex B. Protonolysis of the palladium with generated HX to give the corresponding ketones and regenerate palladium(II) species.

The reaction mechanism claimed by the authors is speculative, and their results are insufficient to explain their proposed mechanism. Based on the experimental results, they should exclude other possible conventional mechanisms such as those mentioned above.

Response: Thanks. We understand your concerns but can hardly agree with you.

The reaction pathway raised by reviewer 2 is unreasonable if considering the substrates and conditions in all the sections, including condition optimization, substrate scope screening and control experiments. The reasons are listed as following:

(1) According to reviewer's mechanism, substrates without phenyl substitute on alkene group should give intermediate **A** as a secondary alcohol. This type of intermediate **A** is impossible to undergo beta-C elimination to give your proposed intermediate **B**. In fact, most substrates in our work in Table 2 and 3 are this type of substrates, but always gave the desired products in satisfied yields.

(2) According to reviewer's mechanism, the compound **7** in equations 4 should give the desired product. However, it was not the truth.

(3) As we shown in Table 1 and equations 7, the oxygen played an important role in our reaction. However, the reviewer proposed mechanism is not related to oxygen.

The mechanism in this manuscript is convinced as it was consistent with all the detailed experiment results including Table 1-3, Scheme 2 and DFT calculations. And it was supported by reviewer 1 and 3.

2. They described their working hypothesis using metalacyclohexene in Scheme 1A, and their proposed intermediate is a metalatetrahydropyridazine derivative, not a metalacyclohexene intermediate. This description needs to be more accurate to understand the novelty of this type of metala-electrocyclic reaction, and they should rationally describe the working hypothesis using a suitable intermediate.

Response: Thanks. We have carefully reviewed previous reports of metalla-DA reactions as a double check, and confirmed that the six-membered metallacycle bearing N and C atoms have been reported in the literatures. To make the description more accurate in Scheme 1A, we redrew the figure as below: N was added to display the known metalla-DA reaction and unknown retro-metalla-DA reactions. Thank you very much.

A. Retro-Metalla-Diels-Alder (DA) reaction for C-C bonds cleavage

3. (1) On page 3, line 52 and 53: they should cite references for supporting the contents (2) and (3).

(2) Wrong nomenclature: isopropanol isopropyl alcohol or 2-propanol

Response:

Thank you very much.

(1) The related literatures for supporting the contents (2) and (3) have been added in ref 34 and 35.

(2) We have corrected the 'isopropanol' to '2-propanol' in the revised manuscript.

To the Reviewer 3:

This paper described an interesting C–C bond cleavage protocol through retro-pallada-Diels-Alder pathway. The reaction shows promising application prospect in chemical modification of β,γ -vinyl ketones due to its high site-selectivity and good functional group tolerances. The authors have proposed six-membered palladacycle mechanisms, and have investigated the detailed mechanism by control experiments and DFT calculations.

After carefully checking the DFT calculations in the paper, this reviewer finds that the selected computation methods are appropriate and provides reliable theoretical insights for this complex chemical synthetic experiment. Step-by-step mechanism presented in this paper suggests that the energetically favored path-III, that well interprets the Pd-catalyzed cyclization and ring-opening processes. The promotion effects of CO and O₂ in their experiments have been reasonably explained through these computational results.

In my opinion, the elaborate reaction path network proposed in this paper contributes to get deeper understanding of the retro-pallada-DA reactions and offers valuable reference for future experimental explorations. I recommend acceptance after addressing the following minor comments.

1. The zero energy should be addressed in SI. The relative energies and free energies (at 298.15K) with respect to Im1 are in kcal/mol.

Response: Thanks. The data has been addressed in the revised SI.

2. ΔG_{TS-a} in Figure S1 was relative difference between ΔG of Im-a and TS-a, which is better expressed as $\Delta\Delta G$.

Response: Thanks. The issue of $\Delta\Delta G$ has been revised in Figure S1.

3. Fonts in Table S1 is not consistent.

Response: Thanks. The 'fonts in Table S1' have been adjusted to keep consistent.

REVIEWERS' COMMENTS

Reviewer #1 (Remarks to the Author):

In terms of reaction mechanism, the mechanism proposed by the author is reliable. There are two reasons: a) In the mechanism proposed by reviewer 2, when the intermediate A has no benzene ring, it will occur preferentially β -hydrogen elimination is not carbon elimination, and most substrates of the author are not substituted by benzene ring; b) The author's mechanism can explain why the reaction yield is less than 5% in the absence of oxygen.

In other aspects, the author has revised it according to the requirements of reviewers.

To sum up, it is recommended that the manuscript be published in Nature Communications in the current version.

Reviewer #2 (Remarks to the Author):

Guan and co-workers have revised their manuscript according to the reviewers' comments. This reviewer recommends publication of their result in Nature Communications without alternation.

Reviewer #3 (Remarks to the Author):

All my concerns have been addressed satisfactorily. I suggest publication of this nice work.

Btw, the author's reply to the mechanism suggested by reviewer 2 is reasonable. Since all the β,γ -vinyl ketone substrates in Table 2 do not have substituents at the β position, the intermediate A for these substrates can be supposed to undergo β -H elimination rather than β -carbon elimination. This is not consistent with the experimental observations. In addition, water is not a good nucleophile.

Point-by-point response to the reviewers' comments

Reviewer #1 (Remarks to the Author):

In terms of reaction mechanism, the mechanism proposed by the author is reliable. There are two reasons: a) In the mechanism proposed by reviewer 2, when the intermediate A has no benzene ring, it will occur preferentially β -hydrogen elimination is not carbon elimination, and most substrates of the author are not substituted by benzene ring; b) The author's mechanism can explain why the reaction yield is less than 5% in the absence of oxygen.

In other aspects, the author has revised it according to the requirements of reviewers.

To sum up, it is recommended that the manuscript be published in Nature Communications in the current version.

Response:

Thank you for recommendations. We appreciate your time in processing our manuscript.

Reviewer #2 (Remarks to the Author):

Guan and co-workers have revised their manuscript according to the reviewers' comments. This reviewer recommends publication of their result in Nature Communications without alternation.

Response:

Thank you for recommendations. We appreciate your time in processing our manuscript.

Reviewer #3 (Remarks to the Author):

All my concerns have been addressed satisfactorily. I suggest publication of this nice work.

Btw, the author's reply to the mechanism suggested by reviewer 2 is reasonable. Since all the β , γ -vinyl ketone substrates in Table 2 do not have substituents at the β position, the intermediate A for these substrates can be supposed to undergo β -H elimination rather than β -carbon elimination. This is not consistent with the experimental observations. In addition, water is not a good nucleophile.

Response:

Thank you for recommendations. We appreciate your time in processing our manuscript.